# The Initial Cardiac Tissue Response to Cryopreserved Allogeneic Adipose Tissue-Derived Mesenchymal Stromal Cells in Rats with Chronic Ischemic Cardiomyopathy

**DOI:** 10.3390/ijms222111758

**Published:** 2021-10-29

**Authors:** Bjarke Follin, Cecilie Hoeeg, Lisbeth D. Højgaard, Morten Juhl, Kaya B. Lund, Kristina B. V. Døssing, Simon Bentsen, Ingrid Hunter, Carsten H. Nielsen, Rasmus S. Ripa, Jens Kastrup, Annette Ekblond, Andreas Kjaer

**Affiliations:** 1Cardiology Stem Cell Centre, The Heart Center, Copenhagen University Hospital Rigshospitalet, 2100 Copenhagen, Denmark; cecilie.hoeeg.pedersen@regionh.dk (C.H.); lisbeth.drozd.hoejgaard@regionh.dk (L.D.H.); morten.juhl@regionh.dk (M.J.); kayabruun@hotmail.com (K.B.L.); Jens.Kastrup@regionh.dk (J.K.); Annette.Ekblond@regionh.dk (A.E.); 2Department of Clinical Physiology, Nuclear Medicine and PET & Cluster for Molecular Imaging, Copenhagen University Hospital-Rigshospitalet & Department of Biomedical Sciences, University of Copenhagen, 2200 Copenhagen, Denmark; kbdoessing@sund.ku.dk (K.B.V.D.); simon.bentsen.01@regionh.dk (S.B.); rasmus.ripa@regionh.dk (R.S.R.); andreas.kjaer@regionh.dk (A.K.); 3Department of Immunology and Microbiology, University of Copenhagen, 2200 Copenhagen, Denmark; 4Minerva Imaging, 3650 Oelstykke, Denmark; ihu@minervaimaging.com (I.H.); chn@minervaimaging.com (C.H.N.)

**Keywords:** mesenchymal stem cell, MSC, cell therapy, heart failure, mode of action, macrophage, cardioimmunology

## Abstract

Mesenchymal stromal cells have proven capable of improving cardiac pump function in patients with chronic heart failure, yet little is known about their mode of action. The aim of the study was to investigate the short-term effect of cryopreserved allogeneic rat adipose tissue-derived stromal cells (ASC) on cardiac composition, cellular subpopulations, and gene transcription in a rat model of chronic ischemic cardiomyopathy (ICM). Myocardial infarction (MI) was induced by permanent ligation of the left anterior descending coronary artery. After 6 weeks, the rats were treated with ASCs, saline, or no injection, using echo-guided trans-thoracic intramyocardial injections. The cardiac tissue was subsequently collected for analysis of cellular subpopulations and gene transcription 3 and 7 days after treatment. At day 3, an upregulation of genes associated with angiogenesis were present in the ASC group. On day 7, increases in CCR2^+^ and CD38^+^ macrophages (*p* = 0.047 and *p* = 0.021), as well as in the CD4/CD8 lymphocyte ratio (*p* = 0.021), were found in the ASC group compared to the saline group. This was supported by an upregulation of genes associated with monocytes/macrophages. In conclusion, ASC treatment initiated an immune response involving monocytes/macrophages and T-cells and induced a gene expression pattern associated with angiogenesis and monocyte/macrophage differentiation.

## 1. Introduction

Heart failure affects millions of people worldwide and causes high morbidity and mortality [1,2]. Most of these clinical cases of heart failure are caused by ischemic cardiomyopathy (ICM) [3]. Existing treatments have reduced mortality; however, the increasing prevalence of heart failure has left a large group of symptomatic patients without further treatment options [4].

Through the past two decades, cell therapy has been found to be a potential new treatment option for patients with ischemic heart failure. Treatments using mesenchymal stromal cells (MSCs) are the furthest in clinical translation, where the first phase III study was recently concluded (NCT02032004). However, a great challenge in the field of cardiac MSC therapy is the lack of understanding of the mode of action (MoA). It has been proposed that the cells mainly exert their function through paracrine actions, leading to improved angiogenesis and decreased fibrosis [1,5,6]. However, the path to this is poorly understood and sparsely investigated. It is recognized that only a fraction of the administered cells is present in the heart after 24 h, while clinical studies have found functional effects to last for years [1,5,7,8]. Thus, the cellular interactions leading to the potential regenerative effects most likely occur in the first days following administration. Therefore, the cells immediately affected by the MSCs are likely mediators of an increased downstream cascade of events [9]. Consequently, we believe that the cellular interactions in the early phase after treatment is of utmost importance for understanding MSC MoA.

A recent study by Vagnozzi et al. found the regenerative effect of MSCs to be attributed to sterile inflammation and immune interaction occurring within the first days following treatment. Vagnozzi et al. applied the cells one week after myocardial infarction (MI) [9], which is common in preclinical studies. However, clinical cell therapy has been shown to be effective in the chronic disease state, while the effect has been absent in acute conditions [10]. The MoA could very well be different in chronic conditions, such as ICM or heart failure, where the inflammatory environment is not as extensive as in the acute and subacute MI. Furthermore, the cell products used in animal studies are usually freshly harvested from a culture of metabolically active and dividing cells, while clinical approaches are moving toward applying cryopreserved products for improved feasibility and scalability of the treatment [4,5,11,12]. Thus, to understand what occurs in the patients, both the disease model and the cell product need to resemble the clinical situation as close as possible.

Using a rat model of chronic ICM, our aim was to identify early changes in cardiac cellular subpopulations and transcription in response to a well-characterized and pure cryopreserved allogeneic rat adipose tissue-derived stromal cell (ASC) product.

## 2. Results

### 2.1. Rat ASC Characterization

Rat ASCs were harvested from subcutaneous fat of male Lewis rats. The phenotype of the ASCs was determined by flow cytometry on the basis of the recommendations of the International Society for Cellular Therapy (ISCT) [13,14]. Not all markers are available for rats, which is reflected in the panel (Table 1). The ASCs in passage 1 displayed high expression of stromal markers CD29, CD90, and CD73, while low expression was observed of CD31, CD45, and CD11b/c. This is standard characterization of ASC or MSCs for preclinical studies (Figure 1).

Since the cells were to be injected immediately upon thawing, resembling clinical applications, the parameters for recovery and function were assessed following cryopreservation and thawing. Recovery of the cells was measured in percentages as the number of ASCs during thawing compared to the number frozen. A mean recovery of 76.0 ± 13.9% was observed.

Adherence was measured in percentages as the number of ASCs attached to the bottom of the culture flasks 5 h after seeding compared to the number of live ASCs seeded. The mean adherence of the ASCs was 86.9 ± 4.3%.

The ability of the ASCs to secrete vascular endothelial growth factor (VEGF) was confirmed using ELISA.

The ASCs were deemed negative for aerobic and anaerobic bacteria using the BactAlert system. Furthermore, the products did not contain any rodent pathogens based on an extensive IDEXX IMPACT V PCR profile analysis (IDEXX BioAnalytics, Ludwigsburg, Germany) and was negative for Mycoplasm (Statens Serum Institute, Copenhagen, Denmark).

These results demonstrated that the cryopreserved allogeneic rat ASCs were pure and responsive upon thawing and adhered to criteria for clinical cell products.

### 2.2. Treatment Effect on Cardiac Composition

Female Sprague–Dawley rats (350–400 g, Charles River) were used in this study. The use of male ASCs from Lewis rats rendered the treatment allogeneic with MHC mismatch based on information from the vendor. The animals were acclimatized for two weeks prior to any operations.

At week 0, MI was induced in the animals by performing a permanent ligation of the left anterior descending (LAD) artery. At week 4, the infarct area was assessed using [^18^F]-fluorodeoxyglucose (FDG) positron emission tomography/computed tomography (PET/CT), and animals were randomized to receive saline, ASCs, or no injections (Figure 2). The choice of saline as the placebo group was to keep the design as close to the clinical studies. The treatment was performed in week 6 by minimally invasive echo-guided trans-thoracic intramyocardial injections. Two injections were performed in the peri-infarct region of the mature scar, with visual confirmation of successful injections. Euthanization and tissue collection was performed at 3 or 7 days after treatment. Tissue biopsies were stored for assessment of cardiac transcription by RT^2^ profiler PCR arrays (Qiagen, Copenhagen, Denmark) and regular qPCR as well as Y-chromosome identification by qPCR. The injection method was verified in a pilot experiment using bioluminescence imaging and transfected human ASCs (Appendix A). In addition, 769.7 pg male DNA was detected using Y-chromosome qPCR in tissue samples harvested at day 3 after treatment in one of four rats. The remaining cardiac tissue was immediately processed for flow cytometric analysis of the cardiac cellular composition. All data handling, including setting gates for flow cytometry, was performed blinded to the treatment.

The cardiac composition was investigated in terms of hematopoietic cells (CD45^+^), endothelial cells (CD31^+^ CD45^−^), and stromal cells (CD90^+^CD45^−^) as percentages of live cells. Four multicolor panels were created to investigate the different cells types, with emphasis on the immune cell compartment. All panels were compensated using CompBeads (BD Biosciences), and gates were set based on fluorescence minus one (FMO).

The non-injected ICM group contained roughly 9% CD45^+^, 68% CD31^+^CD45^−^, and 9% CD90^+^ CD45^−^ cells (Figure 3). This is very similar to the percentages reported by Pinto et al. in their seminal paper on the cardiac composition [15]. The non-injected ICM hearts contained 8.7 ± 3.1% CD45^+^ cells, while the ASC and saline group at 3 days after injection were increased (14.1 ± 3.2% and 16.3 ± 9.6%, respectively). At day 7, the percentages of CD45^+^ cells in the ASC and saline groups (9.2 ± 2.4% and 6.9 ± 0.5%, respectively) were comparable to the ICM. The fact that the increase in the CD45^+^ cells population was only observed at day 3, and in both the ASC and saline groups, suggests that this was due to the immediate inflammatory response to the injection itself. The timing of the response seems similar to other cardiac wound-healing timelines, e.g., after myocardial infarction. Despite this increase in CD45^+^ cells, the CD90^+^CD45^−^ and CD31^+^CD45^−^ cell populations were stable across groups. Interestingly, the percentage of CD31^+^CD45^−^ cells decreased in the ASC group 7 days after injection to a level significantly lower than the saline group (*p* = 0.048). The unchanged CD90^+^CD45^−^ cell population suggests that the injection did not affect the stromal cell population within the first week.

The fact that the changes in CD45^+^ and CD31^+^CD45^−^ did not affect the other populations within the same group points toward changes in the CD31^−^CD45^−^CD90^−^ uncharacterized population. This population was calculated from the contribution of two panels and assumes that the CD31^+^CD45^−^ and CD90^+^CD45^−^ populations do not overlap. The contribution of this uncharacterized population to the total live cells was lowest in saline at day 3 but increased to slightly above ICM levels at day 7. In the ASC groups, the population was similar to ICM levels at day 3 while increasing almost three-fold to 36.2 ± 12.9% at day 7. The level was higher in the ASC group at both day 3 and day 7, although only significant at day 3 (*p* = 0.031). This suggests that the regular wound healing from the injection itself results in an increased fraction of CD31^−^CD45^−^CD90^−^ at 7 days after injection, and that the ASC treatment tended to increase this population more than saline. Our prior experience with this gating would result in the inclusion of some cardiomyocytes in this population. However, with the close resemblance to the data from Pinto et al., which was generated on cardiac non-myocyte cells, CD90^−^ fibroblasts and mesenchymal cells might be the dominating cell types within the population [15]. This was an unexpected finding, which should be explored further in future studies. While some of the answers to ASC MoA may be within this CD31^−^CD45^−^CD90^−^ population, we further investigated the CD45^+^ population.

### 2.3. Monocyte/Macrophage Response to Treatment

Within the CD45^+^ compartment, we investigated monocytes/macrophages (CD11b^+^) and lymphocytes (CD3^+^). To assess the macrophage response, a panel including CCR2, CD38, and CD163 was used. Subpopulations were analyzed with inflammatory markers set against anti-inflammatory markers, such that CCR2 was plotted against CD163 (Figure 4) [16].

As with CD45^+^, there was a tendency toward a higher percentage of CD11b^+^ cells in both the ASC and saline group at day 3 compared to the ICM group and the injection group at day 7. However, this increase was not significant. When investigating the subpopulations, the percentages of CD11b^+^CCR2^+^ monocytes were similar in the ASC and saline groups at day 3. However, while the fraction dropped to ICM levels in the saline group, it remained significantly elevated in the ASC group at day 7 (*p* = 0.047). CCR2^+^ monocytes/macrophages are characterized as pro-inflammatory. They have been found to be responsible for the recruitment of monocytes and neutrophils as well as implicated in pathological post-infarct remodeling [17,18]. The higher fraction of CCR2^+^ cells as in the ASC group at day 7 is most likely a sign of sterile inflammation as an immune response to cell therapy [9].

To further explore a potential inflammatory reaction, we investigated CD38^+^ cells. Approximately 50% of the CD11b^+^ cells were positive for CD38 in all groups except in the ASC group at day 7, where it was 68.8 ± 9.5%. This was higher than all other groups and timepoints and significantly higher than the saline group at day 7 (*p* = 0.016). CD38 was included in the panel, since it has been shown to be highly expressed on pro-inflammatory macrophages in vitro and the best separator between pro- and anti-inflammatory subtypes [19,20]. While in vivo and in vitro markers are not interchangeable, CD38 has also been implicated in pro-inflammatory conditions in vivo. Therefore, the CD38^+^ increase 7 days after ASC treatment confirms that it is a specific inflammatory in response to the ASC treatment. This is consistent with the higher CD11b^+^CCR2^+^ population.

There was no significant difference in CD163^+^ cells across groups. CD163 is a widely used marker of anti-inflammatory phenotype monocyte/macrophages in rats. With this being the only marker for anti-inflammatory action, the results suggest a largely inflammatory response in the myeloid compartment to ASC treatment.

### 2.4. T-Cell Response

To investigate the lymphocytic response to ASC treatment, the fractions of CD3, CD4, and CD8 positive cells were assessed (Figure 5). The cytotoxic CD8^+^ T-cells are responsible for clearing foreign objects, such as allogeneic cells, whereas the CD4^+^ cells can participate in wound healing and scar formation [21,22].

There was no difference in CD3^+^ percentages across groups. This means that the peak in CD45^+^ cells observed at day 3, in both the saline and ASC groups, is due to non-lymphocyte populations, which was also demonstrated with the increased myeloid CD11b^+^ population. The fractions of CD4^+^ and CD8^+^ T-cells were similar across groups, except for the ASC group at day 7, where the fraction of CD4^+^ T-cells increased, while the CD8^+^ T-cells decreased significantly compared to saline (*p* = 0.012 and 0.019, respectively). As a result, the CD4/CD8 ratio increased to 1.14 ± 0.20 in the ASC group compared to 0.66 ± 0.23 in the saline group (*p* = 0.021).

A high CD4/CD8 ratio has been associated with a good prognosis after MI and is frequently used as a measure of adaptive immune function [23,24]. In human and mice, a ratio between 1.5 and 2.5 is considered normal, whereas a ratio < 1 is an indicator of immune impairment and chronic inflammation [24]. Based on this, the regenerative effects of ASC therapy may be partially mediated through modulation of the adaptive immune response.

### 2.5. Treatment Effect on Gene Transcription in the Heart

To assess the effect of the treatment on the gene transcription in the heart tissue, a custom RT^2^ qPCR array was created. The 84 genes investigated were selected based on which genes were specifically related to cardiac regeneration in the literature (Appendix A). Significant differences in expression of a certain gene were subsequently verified by regular qPCR.

At day 3 after treatment, Collagen Type I Alpha 2 Chain (Col1a2), Collagen Type 3 Alpha 1 Chain, Neuregulin 1, Angiopoietin 2 (Angpt2), and Latent Transforming Growth Factor Beta Binding Protein 1 (Ltbp1) were significantly upregulated in the ASC group compared to the ICM group (Figure 6A). According to Gene Ontology, this combination of genes is mainly associated with angiogenesis [25,26,27]. When comparing saline to ICM, only Chemokine (C-C motif) ligand 3 (Ccl3) was significantly upregulated. With the data generated using the RT^2^ arrays, there was no significant difference in transcription between the ASC and the saline group at 3 days. These findings were further investigated using regular qPCR with the same primers. Ltbp1 and Angpt2 was found to be upregulated in the ASC group compared to both ICM and saline, and Ccl3 was upregulated in both the saline and ASC group compared to ICM (Figure 6B).

At day 7 after the treatment, Colony-Stimulating Factor 2, Vegfa, and Vegfb were significantly upregulated in the ASC group compared to ICM (Figure 6C). These changes are associated with chemotaxis (positive chemotaxis (GO:0050918)), monocyte and macrophage differentiation (monocyte differentiation (GO:0030224), and macrophage differentiation (GO:0030225)), as well as angiogenesis (coronary vasculature development (GO:0060976) and sprouting angiogenesis (GO:0002040). When comparing the saline group to ICM, Angpt 1, Integrin Subunit Alpha V, Kinase Insert Domain Receptor, Myosin Heavy Chain 6, Platelet-Derived Growth Factor Subunit A, Platelet And Endothelial Cell Adhesion Molecule 1, Mothers against decapentaplegic homolog 2 (Smad2), Vegfa, and Vegfb were upregulated (Appendix A). This points toward general cellular response to chemical stimulus (cellular response to chemical stimulus (GO:0070887)) and immune cell chemotaxis (cell chemotaxis (GO:0060326), positive regulation of mast cell chemotaxis (GO:0060754), and basophil chemotaxis (GO:0002575)). This suggests that the injection itself induces changes, and that some changes are similar in the ASC and saline groups. When comparing the ASC and saline groups, interleukin 11 and C-X-C chemokine receptor type 4 was upregulated, while Angpt1, Tissue Inhibitor of Metalloproteinases 3 (Timp3), and Smad2 were downregulated in the ASC group compared with the saline group (Figure 6D). According to Gene Ontology, these genes are mostly involved with macrophage migration and positive signal transduction, indicating differences in macrophage responses between the groups. Thus, the effect of ASC on the transcription in the peri-infarct region of the cardiac tissue seems to be related primarily to monocyte/macrophage regulation. However, it is important to note that the enrichment analysis was performed using relatively few genes, and that future studies investigating ASC/MSC MoA should include analysis of macrophage migration.

## 3. Discussion

The MoA of MSC therapy for cardiac regeneration has been extensively discussed for years [1,27,28,29]. It is crucial to understand in order to improve cell therapy, but also for legislative purposes, and for the creation of potency assays investigating the proper in vivo MoA. While many studies find increased angiogenesis and decreased fibrosis, the cellular interactions precluding this effect have been largely overlooked, and this might be the most important interaction in terms of MoA. In addition, most data on the subject are derived from animal models of acute or subacute MI and with freshly harvested cell products, which does not reflect the ongoing clinical development. Therefore, we designed this study to resemble the clinical setting as closely as possible with a well-characterized ASC product following clinical guides for ATMPs.

Our results on cardiac composition resemble those of Pinto et al. The major difference lies within the CD3^+^ cell population. Pinto et al. described that the CD3^+^ cell population constituted 0.3% of the non-myocyte cardiac cell populations in healthy hearts, whereas the present study found this cell population to make up 2.45% in the ICM hearts [15].

Both the data from cellular composition and transcription point toward the local trauma of the injection obscuring potential changes between ASCs and saline at 3 days after treatment. This was suggested by the equal increase in CD45^+^ cells and CD11b^+^ cells, as well as similar tendencies in transcription (Figure 7). However, the initial response from the injection trauma seems to be dampened from day 3 to day 7, which is consistent with regular myocardial wound healing after injury, such as MI [30]. This lowers the background noise from the injection itself and makes it possible to detect differences in cardiac composition and transcription. This timeline after treatment or ischemic insult is similar to findings from other studies [9,22].

Regarding the cardiac cell populations, it was clear that the ASCs affected the immune response differently than the saline at day 7 after treatment, as ASC treatment resulted in a higher percentage of CD11b^+^/CCR2^+^ cells. Furthermore, the fraction of CD11b^+^ cells that were positive for CD38 increased at day 7, with no accompanying increase in CD163^+^ cells. These findings were unexpected, since CCR2 and CD38 are most often considered markers for pro-inflammatory macrophages. As in our study, Vagnozzi et al. found cell treatment to result in an increase in CCR2^+^ cells but also an increase in the anti-inflammatory CXC3R1^+^ cells [9]. This could suggest different cells responding to the therapies in subacute MI and ICM or attest to the difference between CD163 and CXC3CR1 markers for anti-inflammatory phenotypes.

Looking into other indications, Takeda et al. found CCR2^+^ cells to be imperative for the immunosuppressive effect of MSCs in allergic airway inflammation [31]. They found CCR2^+^ cells to cluster around injected MSCs at timepoints close to administration, and the effect of MSCs was abolished when blocking either the CCR2-ligand CCL2 in MSCs or CCR2 systemically. Based on this, the presented hypothesis was that the recruited CCR2^+^ monocytes/macrophages subsequently increased the number of regulatory T-cells (Tregs). In line with this, Whelan et al. [32] found CCL2 to be necessary in MSCs for accelerated excisional wound healing, highlighting this as a very important MSC–macrophage interaction.

Interestingly, our investigation of the cardiac T-cell population found an increase in the CD4/CD8 ratio at day 7, which may have several explanations. Firstly, we expected the CD8^+^ population to increase in the ASC group, as effector cells would be activated to remove the allogeneic ASCs, especially since these are expanded in medium containing FBS [21]. However, this was not the case. In congestive heart failure, both CD4^+^ and CD8^+^ T-cells are expanded and can both add to deterioration of the wound and wound healing [33]. Akiyama et al. [34] found that the MSC-mediated immunoregulation is initiated by T-cell debris leading to activated phagocytes, which in turn differentiated naïve T-cells toward CD4^+^ Treg-cells. This could indeed improve the outcome of the therapy, but our T-cell panel was not set up to investigate Treg populations.

The effect of ASCs on the uncharacterized CD31^−^/CD45^−^ population was interesting. It is intriguing that such a population responds to treatment, suggesting that it might be implicated in the ASC MoA. Consequently, this population should be further investigated and characterized in future preclinical studies. However, transcriptional results are derived from whole cardiac tissue and therefore include all cell populations. According to these data, the most notable difference between ASC and saline, in the peri-infarct tissue, is related to monocyte/macrophage responses. This suggests that it might indeed be the CD45^+^ compartment which is responsible for the initiation of the ASC MoA, despite their lower numbers compared to the other cell compartments.

Most similar studies treat with cell products which are freshly harvested from an ongoing culture. In this way, they investigate the action of viable and metabolically active cells. However, these are not the products that are being used in the clinical setting; thus, the observed results might not mirror the actual response in the patients. By administering cryopreserved cells in their cryogenic medium, we not only investigate the action of the cells but of the whole cell product, which includes the product excipient. The tissue effect could vary with different cryogenic media, but it was not within the scope of this study to investigate this aspect. Despite this, it is the author’s belief that using cryopreserved MSCs is the most clinically relevant approach and thus should be further explored in future studies.

Overall, we found the response after ASC administration to be almost similar to the response to saline at day 3. However, on day 7, the ASC treatment resulted in altered inflammatory monocyte/macrophage subpopulations, an increased CD4/CD8 ratio, and an increased uncharacterized CD31^−^CD45^−^CD90^−^ population. This, together with peri-infarct transcription being associated with macrophage migration, points toward the interaction with the inflammatory macrophages being the dominant initial response after treatment with ASCs in ICM hearts. Further studies will be needed to investigate this response for a more complete understanding of ASC MoA in ICM and heart failure.

## 4. Materials and Methods

### 4.1. Isolation and Culture of Rat ASCs

Lipoaspirate was obtained through surgical resection of subcutaneous fat from 4 male Lewis rats. The lipoaspirate was washed in phosphate-buffered saline (PBS) without Ca^2+^ and Mg^2+^ (Gibco). A collagenase NB 4 (Nordmark Biochemicals, Uetersen, Germany) solution was prepared (0.6 PZ U/mL) by dissolving collagenase in Hank’s Balanced Saline Solution containing CaCl_2_ and MgCl_2_ (Gibco, ThermoFisher Scientific, Roskilde, Denmark). The collagenase solution was added to the adipose tissue 1:1 and incubated at 37 °C under rotation in a mini incubator (Labnet, In Vitro, Edison, NJ, USA) for 45 min. For the isolation of mononuclear cells (MNCs), the tissue suspension was filtered through a 100 μm mesh (BD-falcon), whereupon MNCs were isolated by centrifugation. MNCs were eventually resuspended in Minimum Essential Medium α (αMEM), 10% fetal bovine serum (FBS), and 100 U/mL penicillin/100 μg/mL streptomycin (P/S) (Gibco). The choice of FBS as expansion supplement was made after comparing it with GMP-compliant supplements on rat ASCs. We found that the rat ASCs did not respond to the GMP compliant supplement in the same way as the human ASCs. Afterwards, MNCs were seeded in T75 flasks (Nunc, ThermoFisher Scientific, Roskilde, Denmark) in a density of 4.5 × 10^6^ cells/flask and incubated at 37 °C, 20% O_2_, and 5% CO_2_. When confluent, cells were detached with TrypLE (Gibco) and portioned in cryo tube vials of 1 × 10^6^ in CryoStor^®^ CS10 (BioLife Solutions, Bothell, WA, USA) and stored in liquid N_2_.

For the expansion of ASCs, cells in P0 were thawed and seeded in T75 flasks at a density of 1 × 10^6^ cells/flask. The cells were cultured until confluency, after which they were portioned in cryo tube vials of 3 × 10^6^ P1 ASCs in CryoStor^®^ and frozen until usage.

### 4.2. Evaluation of Rat ASC Phenotype and Quality

The phenotype of the ASCs was determined by flow cytometry [13]. For the analysis, 1 × 10^6^ cryopreserved ASCs were used. The cells were thawed and washed twice in PBS and then resuspended in PBS at a concentration of 1 × 10^6^ cells per mL. For the staining, cells were labeled with Fixable Viability Stain 780 (FVS-780, BD Biosciences, Kongens Lyngby, Denmark) per mL cell suspension and subsequently washed in fluorescence-activated cell sorter (FACS)-PBS, consisting of 1% EDTA (Pharmacy of the Capital Region) and 10% FBS (Life Technology, Carlsbad, CA, USA.) in PBS (Pharmacy of the Capital Region). Finally, cells were labeled with 5 μL antibody/100 cell suspension (Table 1), and data were acquired using flow cytometry (BD FACSLyric, BD Biosciences). Data analysis was performed using FlowLogic (Inivai Technologies, Hørsholm, Denmark) based on a minimum of 20,000 live single cells.

To evaluate ASC quality, their ability to recover, adhere, and secrete VEGF upon thawing was measured. Recovery was measured by counting the cells using a hemocytometer (Neubauer) after thawing and comparing the number to the amount frozen. Adherence was analyzed by thawing ASCs, after which they were seeded in T75 flasks (Nunc, ThermoScientific) and cultured in αMEM containing 10% FBS and 1% P/S for 5 h at 37 °C, 20% O_2_, and 5% CO_2_. Following incubation, ASCs were washed to remove non-adherent cells, after which adherent cells were detached using TrypLE (Gibco). Eventually, the number of ASCs seeded were compared to the number of adherent cells at 5 h.

The ability of ASCs to secrete VEGF was measured using ELISA. ASCs were thawed and seeded in a density of 256,000/well in a nuclon delta surface 96-well plate (Thermo Scientific) containing αMEM containing 10% FBS and 1% P/S. One 96-well plate was cultured under normoxic conditions (37 °C, 20% O_2_, and 5% CO_2_) and 1 was cultured under hypoxic conditions (37 °C, 3% O_2_, and 5% CO_2_). After 72 h, the well plates were centrifuged, and supernatants were collected. VEGF concentrations were measured using a Rat VEGF Quantikine ELISA Kit (R&D systems) according to the manufacturer’s instructions, and concentrations were read on a Fluostar Omega plate reader (BMG Labtech, Ortenberg, Germany).

### 4.3. Infarct Induction

A total of 74 animals were enrolled in this study. Of these, 37 animals survived the MI induction and displayed a clear infarct area on [^18^F]-FDG PET/CT imaging. All animal experiments were approved by the Danish Animal Experiments Inspectorate, authorization number 2016-15-0201-00920. The approval was given 5 July 2016. The infarcts were induced as described in our earlier studies [35,36]. In brief, the animals were anesthetized in 4–5% sevoflurane (Baxter), treated with 0.05 mg/kg buprenorphine (Temgesic, 2care4) subcutaneously and intubated. During mechanical ventilation (UNO microventilator UMV-03, Hallowell EMC), the hair was removed from the chest, and the area was sterilized. A small incision was made in the left part of the thorax. Blunt dissection was performed to loosen the skin, as well as the muscle layers for access to the ribs. The 4th rib was cut, and a retractor was inserted. The pericardium was gently removed, and the left appendage was identified. A permanent ligation using 6–0 Prolene was performed below the left appendage, and ischemia was confirmed by visual discoloration of the ventricle. The ribs and skin were closed using 4–0 vircyl. The animals were treated with buprenorphine in Nutella for the following three days and observed for well-being.

### 4.4. PET/CT Scan and Analysis

For all PET/CT scans, the animals were anesthetized in sevoflurane. Intravenous (IV) access through the tail vein was established, and the hair from the chest and inner thigh was removed prior to placement of electrodes (Comepa Industries, Bagnolet, France). A Siemens Inveon PET/CT scanner (Siemens Medical Solutions, Malvern, PA, USA) was used with a water-heated bed and monitoring of ECG and respiration. For attenuation correction, a 15-min CT scan with full rotation, 360 projections, and 65 kV was performed with 1 mL CT contrast (Omnipaque 350 mg I/mL, Bracco Imaging, Göteborg, Sweden).

[^18^F]-FDG was used to assess infarct area. The animals had been given minimal amounts of food for 5 h prior to the scans in order to deplete the glucose in the blood. During cannulation of the tail vein for IV access, the blood glucose was measured using FreeStyle Precision (MediSense). To decrease circulating free fatty acids and shift the energy consumption of the heart to glucose, 15 mg/kg Acipimox (Sigma Aldrich, St. Louis, MO, USA) was injected subcutaneously 10 min prior to injection of [^18^F]-FDG. A PET scan was performed after a circulation time of 60 min.

PET listmode files were histogrammed into a single time frame before being reconstructed using OSEM3D/OP-MAP reconstruction protocol. Inveon Acquisition Workspace was used to reconstruct PET and CT images, and Inveon Research Workspace (both Siemens, USA) was used for the co-registration of PET and CT images.

The static FDG images were analyzed using Corridor4DM version 2017 (Invia LLC, Ann Arbor, MI, USA). The images were oriented automatically into a short axis and vertical and horizontal long axis. The myocardium was automatically outlined, and the uptake was compared to a normal database created from other Sprague–Dawley rats. The infarcted areas were defined as areas with uptake values lower than 2 SDs from the normal database at the given location. The data output was infarct area percentage of total left ventricle as a summation of the AHA 17 segments model. All orientations and outlines were reviewed by experienced observers blinded to the treatment.

### 4.5. Treatment

Treatment was performed using the Vevo 3100 Preclinical Imaging Platform. Prior to ASC or saline treatment, animals were anesthetized, hair was removed as mentioned previously, and analgesic treatment with buprenorphine was administered subcutaneously. A vial of 3 × 10^6^ cryopreserved ASCs were thawed at room temperature; then, they were centrifuged and resuspended to reach a concentration of 1 × 10^7^ ASCs pr. mL CryoStor10 in the syringe. The injection needle was aligned with the transducer, after which the heart was localized in short axis view. The 30 G needle (BD Microlance, Herlvev, Denmark) was inserted in the myocardium through the intercostal space, with needle penetration of the myocardium being confirmed by 2–3 arrhythmic ECG peaks. Injection was performed anterior and lateral/inferior to the MI and visualized as white microbubbles in the myocardium on the echo image. The needle was retracted, and the animals were observed after the procedure. The operator was blinded to treatment group.

To ensure injection into the myocardium, initial bioluminescence imaging was used to verify the echo-guided method. L2T-transduced human ASCs, which were previously described in an earlier study, were injected into the myocardium of a healthy rat using the above-mentioned method [35]. Then, 24 h after ASC administration, luciferin was injected into the abdominal cavity in a dose of 30 mg/kg. The emitted light from the luciferin–luciferase reaction was detected using an IVIS^®^ Lumina XR (Caliper Lifesciences, PerkinElmer, Hopkinton, MA, USA) and Living Image^®^ software v.4.3.1 (PerkinElmer), and the distribution of cells was assessed. Subsequently, the animals were euthanized, and the hearts were excised and cut into 3 parts to assess the location of the cells in the myocardium (Appendix A).

### 4.6. Isolation and Dissociation of Cardiac Tissue

At day 3 or 7 following treatment, rats were euthanized, and the hearts were extracted and perfused using isotonic saline (Amgros, Copenhagen, Denmark), after which atrias were cut off. For RT^2^ PCR profiler arrays, approximately 30 mg of the peri-infarct zone was stored in RNAlater (Sigma-Aldrich) at −80 °C until usage. The remaining heart tissue was stored in Tissue Storage (MACS Miltenyi, Auburn, CA, USA) on ice until dissociation.

For the dissociation of heart tissue, the tissue was cut into pieces of 1 mm^2^ on a 4 °C surface plate (Medite), whereupon an enzyme mix from a Neonatal Heart Dissociation Kit, mouse, and rat (MACS Miltenyi Biotec) was prepared and added according to the manufacturer’s protocol. The tissue was dissociated under 200 rpm agitation (Heto) at 37 °C for 45 min, after which the cell suspension was diluted in PBS, filtered through a 70 μm filter (Falcon), and centrifuged. Following centrifugation, hemolysis was performed using VersaLyse (Beckman Coulter, Marseille, France) according to the manufacturer’s instructions. Upon hemolysis, the cell suspension was gently filtered through a 70 μm filter and resuspended in PBS.

### 4.7. Flow Cytometry

After the isolation and dissociation of the heart tissue, cells were stained with FVS-780, washed in FACS-PBS, centrifuged, and resuspended in 100 μL FACS-PBS for staining with initially titrated antibodies (Table 2).

Finally, a FACSLyrics (BD Biosciences) flow cytometer was used for acquisition. The protocol was generated with automatic compensation using beads (BD Biosciences) and Fluorescence Minus One Controls. Samples were acquired on a medium flow rate, reaching a maximum event rate of 10,000 events/s until at least 500,000 events were acquired. Debris, dead cells, and doublets were excluded from the final data analysis, using FlowLogic software (Inivai), and gating was performed as illustrated in Figure 2, Figure 3, and Figure 4.

### 4.8. RNA Purification and cDNA Synthesis

Prior to DNA and RNA purification, frozen rat heart tissue was lysed and homogenized using a Precellys Lysing Kit (Bertin Instruments). Up to 30 mg of frozen tissue sample was transferred to a 2 mL Precellys tube containing ceramic beads, after which 600 μL Buffer RLT Plus (Qiagen) and 6 μL 14.3 M β-mercaptoethanol were added. Samples were lysed and homogenized for 20 s using a Precellys 24 Tissue Homogenizer (Bertin Instruments) and then centrifuged briefly, after which a 600 mL lysed sample was used for purification. Total RNA was automatically purified from the lysed sample using an AllPrep^®^ DNA/RNA/miRNA Universal Kit (Qiagen) on a QIAcube (Qiagen), following the manufacturer’s protocol for large samples (Qiagen). RNA purity was quantified using NanoDrop (Thermo Scientific). Synthesis of cDNA was performed on 0.5 µg RNA using the RT^2^ First Strand kit (Qiagen) according to manufacturer’s protocol on a Veriti 96-well fast thermal cycler (Applied Biosystems).

### 4.9. RT^2^ Profiler PCR Arrays

Custom RT^2^ Profiler PCR Arrays (Qiagen) were designed and used for analysis of the 84 most relevant genes associated with cardiac regeneration (Appendix A). The reactions were prepared according to the manufacturer’s instructions and analyzed using a PCR plate reader (Bio-Rad, CFX connect™, Copenhagen, Denmark) and Bio-Rad CFX software. A two-step amplification cycle, with initial denaturation at 95 °C for 10 min, followed by 40 cycles of denaturation at 95 °C for 15 s and annealing and elongation at 60 °C for one minute, was applied. A melting curve analysis was performed to detect primer dimers.

Verification of the RT^2^ profiler results was done using QuantiTech primers (Qiagen) in regular qPCR experiments. The amplification reactions were run in duplicates according to the manufacturer’s instructions. A two-step amplification cycle was performed with initial denaturation at 95 °C for 15 min followed by 40 cycles of denaturation at 94 °C for 15 s, annealing at 55 °C for 30 s, and elongation at 72 °C for 30 s. Finally, a melting curve analysis was performed to detect potential primer dimers.

Annotation was performed on upregulated genes using PANTHER (version 16, Released 24 February 2021) on a Gene Ontology database of biological processes complete (Released 2 July 2021) against the reference list on Rattus norvegicus.

### 4.10. Y-Chromosome qPCR

qPCR was performed to detect the expression of the sex-determining region-2 on the Y-chromosome (Sry-2), originating from the injected ASCs (*n* = 4). The amplification reactions were run in duplicates, in a final volume of 25 μL per reaction. Each reaction contained 600 nM of both forward and reverse primer, 50 ng DNA, and SYBR Green Mastermix (Qiagen). Primer sequences were obtained from Puppi et al. (Table 3) [37]. The PCR plate was analyzed using a PCR plate reader (Bio-Rad, CFX connect™) and CFX Maestro Software. A two-step amplification cycle, with initial denaturation at 95 °C for 3 min, followed by 40 cycles of denaturation at 95 °C for 10 s and annealing and elongation at 55 °C for 30 s was applied. A melting curve analysis was performed to detect primer dimers. A standard curve was prepared, using a six-point, four-fold serial dilution series, to allow calculation of the number of remaining ASCs in the myocardium.

### 4.11. Statistics

Statistical analyses were performed in SPSS software version 25, and graphical depiction of data was performed in GraphPad Prism 9 with standard errors. A Shapiro–Wilk test was used to test the normality of data and Levene’s test was used to assess the homogeneity of variance. Data on cell populations from the ASC and saline groups within the same day after treatment were compared. If assumptions were met, Student’s *t*-test was performed, and otherwise, a Kruskal–Wallis test was used. GeneGlobe (Qiagen) was used to analyze the RT2 Profiler PCR Array data. Cq values of target genes were normalized to the geometric mean of the included reference genes. Then, differences in expression levels were calculated using an independent samples *t*-test of 2-ΔCq values between the groups. For all statistical analyses, a *p*-value < 0.05 was considered statistically significant. Results are expressed as mean ± standard deviation (SD).

## Figures and Tables

**Figure 1 ijms-22-11758-f001:**
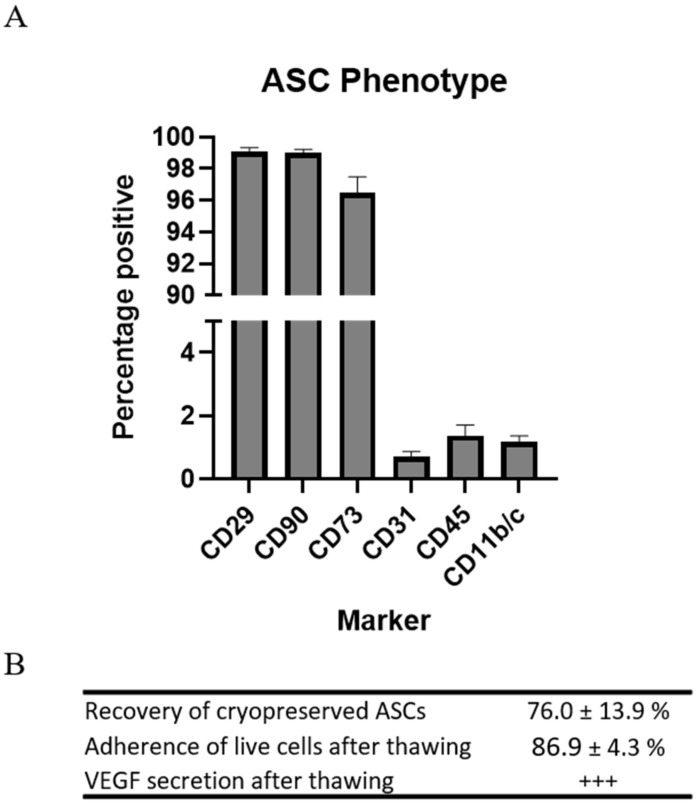
Characterization of preclinical ASCs. (**A**) Regular MSC phenotyping using flow cytometry. *n* = 4. (**B**) Assessment of recovery, test of adherence, and VEGF secretion. *n* = 2.

**Figure 2 ijms-22-11758-f002:**
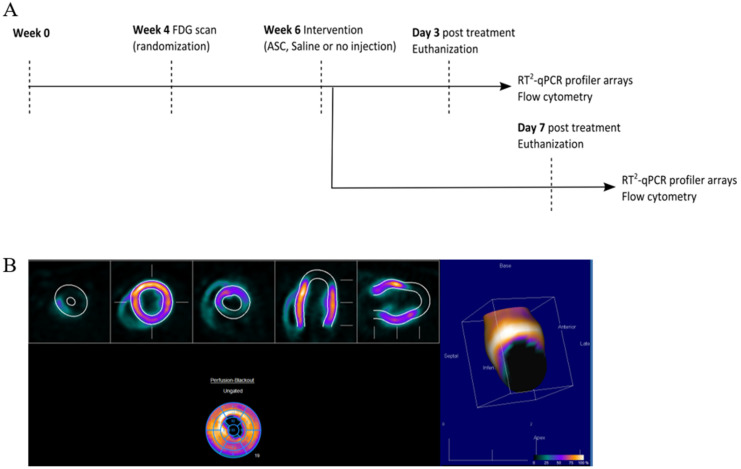
Study design and analysis of cardiac cell composition. (**A**) Study outline. (**B**) FDG PET/CT imaging quantification for infarct assessment.

**Figure 3 ijms-22-11758-f003:**
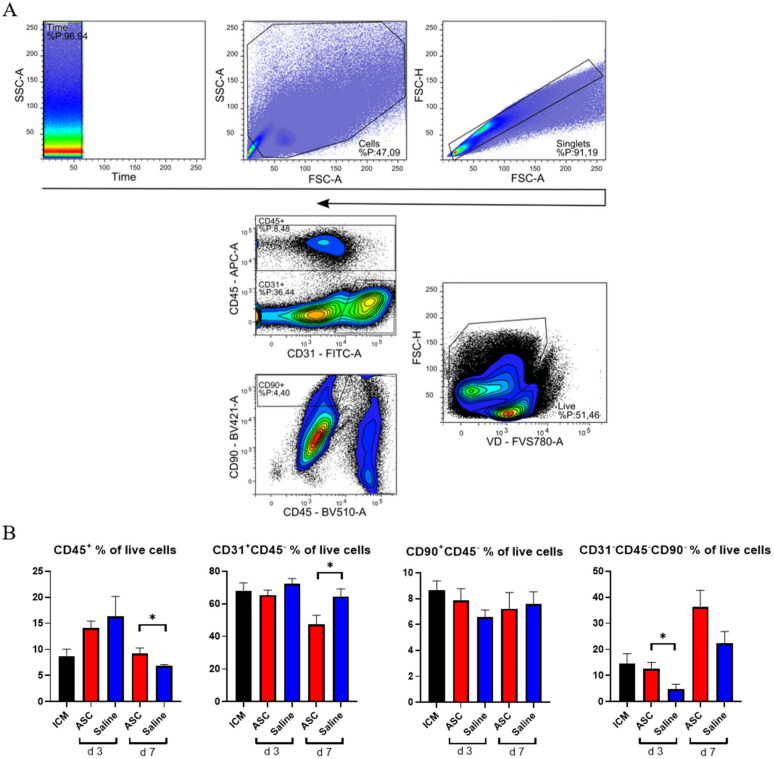
Analysis of cardiac cell composition. (**A**) Gating strategy for the cardiac composition for CD31, CD45, and CD90 across panels. (**B**) Cardiac composition results are depicted for CD45^+^ cells, CD31^+^CD45^−^ cells, CD90^+^CD45^−^, and an uncharacterized population of CD31^−^CD45^−^CD90^−^. ICM *n* = 5, ASC day 3 *n* = 6, saline day 3 *n* = 6, ASC day 7 *n* = 5, and saline day 7 *n* = 5. ICM; ischemic cardiomyopathy. * *p* < 0.05.

**Figure 4 ijms-22-11758-f004:**
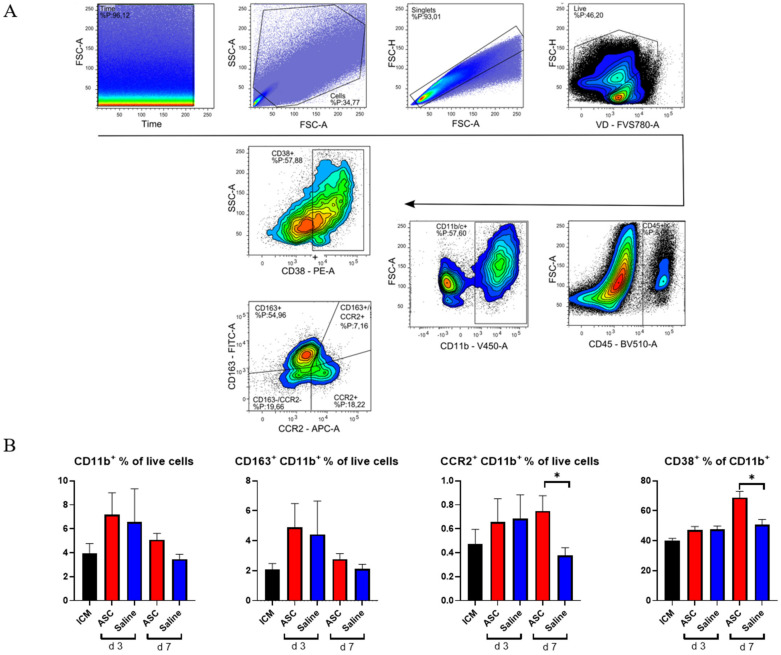
Myeloid response to ASC and saline administration. (**A**) Gating strategy for analysis with double plot of CCR2 and CD163. (**B**) Effect on CD11b^+^ cells, CD163^+^ cells, CCR2^+^ cells, and CD38^+^ cells. ICM *n* = 5, ASC day 3 *n* = 6, saline day 3 *n* = 6, ASC day 7 *n* = 5, and saline day 7 *n* = 4. ICM; ischemic cardiomyopathy. * *p* < 0.05.

**Figure 5 ijms-22-11758-f005:**
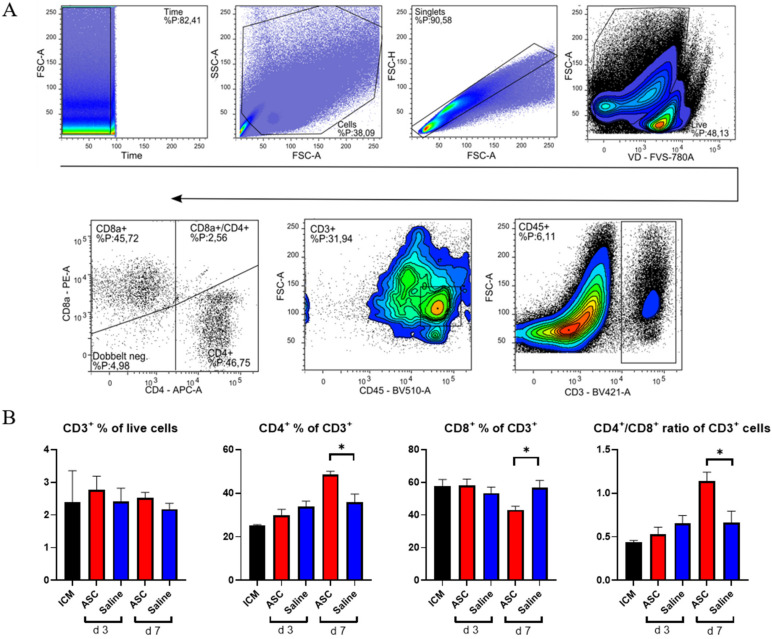
T-lymphocyte response to ASC and saline administration. (**A**) Gating for T-lymphocytes. (**B**) Effect on CD3^+^ cells, CD4^+^, CD8^+^, and the CD4^+^/CD8^+^ ratio. ICM *n* = 3, ASC day 3 *n* = 4, saline day 3 *n* = 4, ASC day 7 *n* = 5, and saline day 7 *n* = 4. ICM; ischemic cardiomyopathy. * *p* < 0.05.

**Figure 6 ijms-22-11758-f006:**
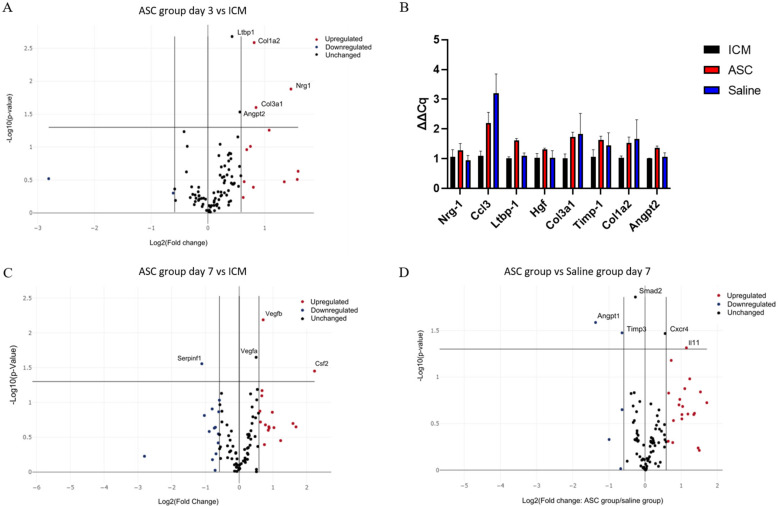
Effect of ASC treatment and saline administration on peri-infarct transcription. (**A**) Volcano plot of ASC vs. ICM hearts at day 3 after treatment. *n* = 5, 5. (**B**) Verifying qPCR of targets between ASC and saline groups at day 3 after treatment. Data are presented as ∆Cq values calibrated to average of the ICM group. *n* = 3, 3, 4. (**C**) Volcano plots for comparison 7 days after treatment between ICM hearts and ASC treatment (*n* = 5 and 4, respectively), or (**D**) ASC treatment vs. saline administration at day 7 (*n* = 4 and 3, respectively). ICM; ischemic cardiomyopathy.

**Figure 7 ijms-22-11758-f007:**
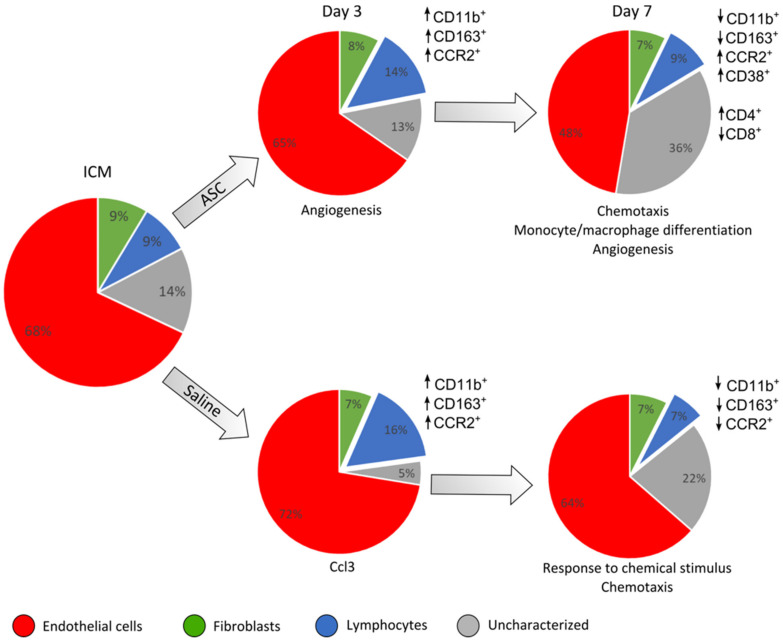
Summary of the findings of changes in cardiac cell composition and peri-infarct transcription. This is based on the results depicted in previous figures. The cell populations based on CD45, CD31, and CD90 are depicted in pie charts for each group and day after treatment. The response in CD45+ cell subpopulations compared to ICM is depicted for the different markers besides the charts. The interpretation of transcriptional results from the peri-infarct region is noted below the charts. ↑ refers to an increased percentage of the given cell population and ↓ a decreased percentage.

**Table 1 ijms-22-11758-t001:** Antibodies used for ISCT characterization, including fluorochromes, clones, and manufacturer.

Antibody	Fluorochrome	Clones	Manufacturer	Concentration
CD11b/c	PE	OX-43	BD Biosciences	1:20
CD29	BV421	Ha2/5	BD Biosciences	1:20
CD31	PE	TLD-3A12	Bio-Rad	1:20
CD45	PE	OX-1	BioLegend	1:20
CD73	Unconjugated	5F/B9	BD Biosciences	1:20
Goat-anti-mouse IgG	FITC		R&D Systems	1:20
CD90	BV421	OX-7	BD Biosciences	1:20

**Table 2 ijms-22-11758-t002:** Descriptions of panels including antibodies, fluorochromes, clones, manufacturer, and staining concentration.

Panel	Antibodies	Fluorochrome	Clones	Manufacturer	Concentration
Macrophage panel	CD11b	V450	WT.5	BD Biosciences	1:50
	CD45	BV510	OX-1	BD Biosciences	1:100
	CD163	FITC	ED2	Bio-Rad	1:50
	CD38	PE	14.27	BioLegend	1:100
	CD86	BB700	24F	BD Biosciences	1:100
	CCR2	APC	890231	R&D Systems	1:25
Lymphocyte panel	CD3	BV421	1F4	BD Biosciences	1:25
	CD45	BV510	OX-1	BD Biosciences	1:100
	CD45RA	FITC	OX-33	BD Biosciences	1:100
	CD8a	PE	OX-8	BD Biosciences	1:50
	CD4	APC	OX-35	BD Biosciences	1:50
Endothelial panel	CD31	BB515	TLD-3A12	BD Biosciences	1:200
	CD45	AF647	OX-1	eBioscience	1:100
Stromal cell panel	CD90	BV421	OX-7	BD Biosciences	1:100
	CD45	BV510	OX-1	BD Biosciences	1:100

**Table 3 ijms-22-11758-t003:** Primer sequences.

Gene	Forward Primer Sequence	Reverse Primer Sequence
SRY-2	5′-CAT CGA AGG GTT AAA GTG CCA-3′	5′-ATA GTG TGT AGG TTG TTG TCC-3′C

## Data Availability

Data generated in the study is available at https://doi.org/10.7910/DVN/REJTRX.

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
