# Peer review of "The Initial Cardiac Tissue Response to Cryopreserved Allogeneic Adipose Tissue-Derived Mesenchymal Stromal Cells in Rats with Chronic Ischemic Cardiomyopathy"

_ijms, 2021, doi:10.3390/ijms222111758_

Round 1

Reviewer 1 Report

Folling et al. investigate the effect of cryopreserved allogeneic rat adipose tissue-derived stromal cells (ASC) on cardiac cellular subpopulations and on gene transcription after their administration in a rat model of chronic ischemic cardiomyopathy (ICM). The aim of this study was to understand the mode of action of ASC. However, some parts of the manuscript, particularly the results, are unclear and end up in overinterpretation.

Major comments:

- The title is confusing and difficult to understand.

- The writing of the aim (lines 70-72) is unclear.

- The figure legends do not indicate the P values.

- Figure 1B (Assessment of recovery, test of adherence, and VEGF secretion): Data of VEGf secretion on Figure 1 is missing as wel as the comparative data with their humana counterpart thatn the authors mentioned that rat ASCs secreted grater amounts of VEGF under both, normoxic and hypoxic conditions.

- What is the treatment to the cells to be injected after thawing.

- “det products”: correct line 99.

- Lines 115-117: Two injections were performed in the peri-infarct region of the mature scar, with visual confirmation of successful injections: please, explain how you confirm the sucsses of injections by visual confirmation.

- The ICM animals group of the figures were analyzed at day 3 or day 7?

- Figure 2: The authors said that This higher percentage of CD45+ cells was comparable to the ICM hearts in the ASC (9.2 ± 2.4 %) and saline (6.9 ± 0.5 %) groups at day 7. As seen in the figure, th porcentage are identical to the ICM group.

- Lines 141-143: "This higher 141 percentage of CD45+ cells was comparable to the ICM hearts in the ASC (9.2 ± 2.4 %) and saline (6.9 ± 0.5 %) groups at day 7, suggesting that this was due to the immediate inflammatory response to the injection" What is the justification of this rationale?

- Lines 145-147: Interestingly, the percentage of CD31+CD45- cells decreased in the ASC group 7 days after injection to a level significantly lower than the saline group (p = 0.048). What does this "interesting" result suggest? Why the authors compare between ACS and saline and do not compare with ICM group?

- Figure 2: The level of CD31-CD45- uncharacterized population was higher in the ASC group both day 3 and day 7, though only significant at day 3 (p = 0.031). However, this level at day 3 was the same than in the ICM group. What is the relevance of this result?

- Figures 3 and 4: Plots are too small. Please make them larger so the gating strategy can be seen.

- Figure 3: The higher percentages in CD11b+ cells and CD11b+CCR2+ monocytes claim by the authors are not significant and any conclusions about them are overstated. On the other hand, the same not significant percentages are shown for CD163+ cells and the authors conclude than there is not a singnificant difference in CD163+ cells across groups.

- Lines 236-237: According to GeneOntology, this combination of genes is mainly associated with angiogenesis. Please add a bibliographic citation to justify this statement.

- Figure 5: Please write the letters in figure 5 bigger so that they can be read properly. In addition, change the dots of the panels because it is impossible impossible to distinguish between downregulated and unchaged conditions.

The statistical analysis is missing and the authors say that several genes are significantly upregulated in the ASC group compared to ICM at day 3 and 7. (Figure 5).

- Figure 5C and D: Why the comparison at 7 days after treatment between (C) ICM hearts and ASC treatment, or (D) ICM vs saline administration shows the upregulation of different group of genes?

Why the injection of saline induces changes in more genes than the ACS injection? It seems than the saline have more effect than the ASC injection.

- Materials and Methods: Why the authors use confluent cells in PO and P1 to cryopreserve them until use? The phenotype of the ASCs was determined in fresh ASCs or in ASCs after thawing? Please explain how you count the cells for the recovery measurement. Do you count all the cells or live cells? Lines 452-453: The cryopreserved ASCs were thawed at room temperature, then centrifuged and resus-452 pended to reach a concentration of 10 x 106 ASCs pr. ml CryoStor10 in the syringe- Does it means that the cells were resuspended in CryStor10 solution? Why the authors use this medium for resuspension and injection? Line 462: Please explain the L2T transduction of human ASCs. Please, explain Figure S1 and add the figure legend.

Author Response

We thank the reviewer for his/her comments. Below, we have tried to address the comments. 

The title is confusing and difficult to understand.

We have now changed the title to : ”The immediate cardiac tissue response to cryopreserved allogeneic adipose tissue-derived mesenchymal stromal cells in rats with chronic ischemic cardiomyopathy

The writing of the aim (lines 70-72) is unclear.

The aim has been rewritten and is now phrased as follows:

Using a rat model of chronic ICM, our aim was to identify early changes in cardiac cellular subpopulations and transcription in response to a well characterized and pure cryopreserved allogeneic rat adipose tissue-derived stromal cell (ASC) product.

The figure legends do not indicate the P values.

P values have been added to the figure legends.

Figure 1B (Assessment of recovery, test of adherence, and VEGF secretion): Data of VEGf secretion on Figure 1 is missing as wel as the comparative data with their humana counterpart thatn the authors mentioned that rat ASCs secreted grater amounts of VEGF under both, normoxic and hypoxic conditions.

Our comparative data on human ASCs will be included in a manuscript in the future and therefore cannot be disclosed. We understand that the comparison therefore cannot be proven, and have chosen to simply describe that the ASCs were able to secrete VEGF. 

What is the treatment to the cells to be injected after thawing.

This is described in lines 438-439.

“det products”: correct line 99.

This has been corrected.  

 Lines 115-117: Two injections were performed in the peri-infarct region of the mature scar, with visual confirmation of successful injections: please, explain how you confirm the sucsses of injections by visual confirmation.

This is described in lines 442-444.

The ICM animals group of the figures were analyzed at day 3 or day 7?

The ICM group should be stable at this time point after the initial infarction. They did not receive an injection, and therefore it is not important at which day the analyses were performed. For practical reasons, the ICM hearts were analyzed 3 days after the saline and ASC rats received injections.

Figure 2: The authors said that This higher percentage of CD45+ cells was comparable to the ICM hearts in the ASC (9.2 ± 2.4 %) and saline (6.9 ± 0.5 %) groups at day 7. As seen in the figure, th porcentage are identical to the ICM group.

What we meant was, that there is an increase in CD45+ cells at day 3 in both the ASC and saline group compared to the ICM group, and that this population is lower again at day 7. This was not clear in the previous version of the manuscript. This sentence has been rewritten to the following:

The non-injected ICM hearts contained 8.7 ± 3.1 % CD45+ cells, while the ASC and saline group at 3 days after injection were increased (14.1 ± 3.2 % and 16.3 ± 9.6 %, respectively). At day 7, the percentages of CD45+ cells in the ASC and saline groups (9.2 ± 2.4 % and 6.9 ± 0.5 %, respectively) were comparable to the ICM. The fact that the increase in the CD45+ cells population was only at day 3 in both the ASC and saline groups suggests that this was due to the immediate inflammatory response to the injection itself.”

Lines 141-143: "This higher 141 percentage of CD45+ cells was comparable to the ICM hearts in the ASC (9.2 ± 2.4 %) and saline (6.9 ± 0.5 %) groups at day 7, suggesting that this was due to the immediate inflammatory response to the injection" What is the justification of this rationale?

This is based on the classical myocardial wound healing timeline (after ex. myocardial infarction), which is characterized by an increase in CD45+ cells the first three days, with a subsequent decline towards day 7 (Prabhu and Frangogiannis, 2016, doi: 10.1161/CIRCRESAHA.116.303577).

Lines 145-147: Interestingly, the percentage of CD31+CD45- cells decreased in the ASC group 7 days after injection to a level significantly lower than the saline group (p = 0.048). What does this "interesting" result suggest? Why the authors compare between ACS and saline and do not compare with ICM group?

The saline and ICM group have approximately the same percentage of CD31+CD45- cells. The results are interesting as most studies describe that ASCs lead to increased angiogenesis and sprouting, suggesting the amount of CD31+CD45- cells likewise would increase. However, as we found a reduced percentage of this cell population along with a gene expression associated with angiogenesis, further research should be conducted to elucidate these mechanisms.

We wish to investigate the difference between injecting saline and allogeneic cryopreserved ASCs, and therefore the comparisons are made between these groups within the different timepoints. This is because the ongoing clinical studies use saline as control, and the current design is aimed at elucidating the MoA differences between the two treatment arms.

Figure 2: The level of CD31-CD45- uncharacterized population was higher in the ASC group both day 3 and day 7, though only significant at day 3 (p = 0.031). However, this level at day 3 was the same than in the ICM group. What is the relevance of this result?

It is very relevant that there is an uncharacterized population which responds to injection/treatment. This population seems to be decreased by an injection in itself (saline) while ASCs retain the population at day 3. The population subsequently increases in both groups and is larger in the ASC group at day 7. This population may be key in the mode of action of ASC in ICM. Unfortunately, our flow cytometry panels were not set up to investigate this population, and further investigations should be made to investigate the population further.

Figures 3 and 4: Plots are too small. Please make them larger so the gating strategy can be seen.

The plots have been changed in order to make the gating strategy more readable

Figure 3: The higher percentages in CD11b+ cells and CD11b+CCR2+ monocytes claim by the authors are not significant and any conclusions about them are overstated. On the other hand, the same not significant percentages are shown for CD163+ cells and the authors conclude than there is not a singnificant difference in CD163+ cells across groups.

As for CD11b+ it is now written that this increase was not significant. However, regarding the CD11b+CCR2+ cell population, this populations in the ASC group was significantly higher than the saline at day 7, as described in line 176, thus we believe that commenting on this result is appropriate.

Lines 236-237: According to GeneOntology, this combination of genes is mainly associated with angiogenesis. Please add a bibliographic citation to justify this statement.

GeneOntology is a large bioinformatics database that builds on extensive libraries. When entering the genes that are significantly upregulated, GeneOntology provides the biological mechanisms that are most likely connected to the specific expression pattern. The database information is currently provided as instructed by GeneOntology (lines 510-512). Additional references (25, 26, 27) about the software have now been added.

Figure 5: Please write the letters in figure 5 bigger so that they can be read properly. In addition, change the dots of the panels because it is impossible impossible to distinguish between downregulated and unchaged conditions.

The colors cannot be changed as these are produced in Qiagens software. The threshold lines in the plot indicate level of significance and fold changes, thus describing which genes are up- or downregulated or unaltered. The size of the font has been altered.

The statistical analysis is missing and the authors say that several genes are significantly upregulated in the ASC group compared to ICM at day 3 and 7. (Figure 5).

The statistics section can be found on lines 257-534, however, more details on the RT2 statistics have been added.

Figure 5C and D: Why the comparison at 7 days after treatment between (C) ICM hearts and ASC treatment, or (D) ICM vs saline administration shows the upregulation of different group of genes?

The different treatments resulted in different transcriptional responses in the tissue. This is consistent with the changes in the cell populations. However, based on feedback from another reviewer, the differences between the ASC and saline group is now highlighted with a volcano plot between these groups at day 7. The volcano plot for saline vs ICM hearts is moved to supplementary.  

Why the injection of saline induces changes in more genes than the ACS injection? It seems than the saline have more effect than the ASC injection.

We agree that this is indeed curious. We cannot explain why this occurs, but only that it does. It will be difficult to go into further detail regarding these results without being too speculative. A future study with deeper transcription analysis will be needed to clarify this.

Why the authors use confluent cells in PO and P1 to cryopreserve them until use?

The cells are cultured until confluency in order to obtain the amount needed for treatment. They are subsequently frozen to mimic novel cryopreserved clinical cell products.

The phenotype of the ASCs was determined in fresh ASCs or in ASCs after thawing?

The phenotype was determined on frozen ASCs. This has now been added to the text (lines 366-367).

Please explain how you count the cells for the recovery measurement. Do you count all the cells or live cells?

The cells were counted using a hemocytometer. This has now been added to the text (lines 377-378)

Lines 452-453: The cryopreserved ASCs were thawed at room temperature, then centrifuged and resus-452 pended to reach a concentration of 10 x 106 ASCs pr. ml CryoStor10 in the syringe- Does it means that the cells were resuspended in CryStor10 solution? Why the authors use this medium for resuspension and injection?

Yes, the cells were resuspended in Cryostor10. The reason for this is that clinical cell products often are administered in their cryogenic medium e.g. Cryostor10.

Line 462: Please explain the L2T transduction of human ASCs.

The L2T transduction was performed using lentiviral transduction as described in our previous work (Follin et al. 2018, doi: 10.1155/2018/7821461). This reference is now inserted in the methods section.

Please, explain Figure S1 and add the figure legend.

The figure has now been described in the legend. The figure heading has also been changed.

Reviewer 2 Report

Distinguished Authors,

I carefully read the article ”Effect of cryopreserved allogeneic adipose tissue-derived mesenchymal stromal cells on cardiac cell composition and transcription in rats with chronic ischemic cardiomyopathy.” by Bjarke Follin et al..

The main focus of the work is to define the mode of action by which infused ASC can repair cardiac function after induced ischemic cardiomyopathy.

I think this is an important but highly demanding aim and, in my opinion authors couldn’t fully elucidate mechanism or at least part of them.

Please find below a list of questions and  concerns about the work.

Could author demonstrate a beneficial effect of infused ASC on cardiac function in their actually applied murine model?

Beside IVIS imaging at day 1, was ASC engraftment in the heart evaluated by other means or at extended time points?

Can authors exclude or alternatively hypothesize that the CD31-/CD45- “unidentified” population can derive from engrafted ASC?

Authors suggested a potential proinflammatory action of ASC showing higher CCR2+ monocytes in ASC treated mice. Nevertheless, ASC infusion is mostly known to play an anti-inflammatory role (as e.g. in clinical trials for GvHD in cancer patients). How presently shown results can cope with such previous evidences?

Considering RT results, rather weak results were shown for Ltbp1 and Angpt2 at day 3. Results regarding day 7 are confusingly reported and the crucial comparison ASC vs saline is not reported in figure 5. Moreover, macrophage migration and positive signal transduction “related effects” seem to be too general and no data can support this hypothesis (macrophage infiltration analysis…?)

It is not clear how figure 6 was built and in my opinion it appears as data repetition within the same work. Moreover pie charts are not normally suggested for data comparison.

Thus, in my opinion, a strong conclusion about ASC mode of action in regenerative medicine can’t be obtained from published results.

Moreover, authors suggest that presently used frozen ASC are suitable for cardiac regenerative medicine applications but a direct comparison with fresh ASC administration is lacking and cardiac function benefits linked to such cell therapy application are not presently evident.

Author Response

We thank the reviewer for his/her comments. Below, we have tried to address the comments. 

Could author demonstrate a beneficial effect of infused ASC on cardiac function in their actually applied murine model?

The aim of this study was not to investigate the functional effect of ASC therapy, but to focus on the initial effect in the tissue.

Beside IVIS imaging at day 1, was ASC engraftment in the heart evaluated by other means or at extended time points?

ASC engraftment was also investigated using qPCR. As male ASCs were used for treatment in a female rat model, quantification of Y-chromosome in the heart tissue was used to investigate the presence of ASCs. At day 3 we found evidence of Y-chromosomes in one out of four rats. The rapidly declining retention percentage is acknowledged both in our previous experiments (Follin et al. 2018, doi: 10.1155/2018/7821461) and in the literature (Hou et al. 2005, doi: 10.1161/CIRCULATIONAHA.104.526749, Vrtovec et al. 2013, doi: 10.1161/CIRCULATIONAHA.112.000230, Terrovitis et al. 2009, doi: 10.1016/j.jacc.2009.04.097).

Can authors exclude or alternatively hypothesize that the CD31-/CD45- “unidentified” population can derive from engrafted ASC?

Based on the size of this population, the amount of Y-chromosome detected in the heart and the number of cells injected, it is unlikely that the “unidentified” cell population is derived from ASCs. IS would also suggest that the remaining administered ASCs proliferate at a relatively high rate in the heart, which is not supported by the literature.

Authors suggested a potential proinflammatory action of ASC showing higher CCR2+ monocytes in ASC treated mice. Nevertheless, ASC infusion is mostly known to play an anti-inflammatory role (as e.g. in clinical trials for GvHD in cancer patients). How presently shown results can cope with such previous evidences?

We agree that the most frequently observed effect of MSC therapy is anti-inflammatory. This is usually described as an effect of the anti-inflammatory properties of MSCs creating a shift in the local macrophages or lymphocytes. However, recent evidence within cardiac cell therapy indicate that the initial response is a sterile inflammatory response, which subsequently leads to an anti-inflammatory effect (Vagnozzi et al, doi: 10.1038/s41586-019-1802-2). This is consistent with data from other indications, which could be ascribed to the anti-inflammatory effect being a result of a cascade initiated by sterile inflammation. The hypothesis is, that the initial immune response triggers the anti-inflammatory response. We sought to elucidate the first response in this process, since we believe that this is the actual mode of action of the ASCs, with downstream effects being a consequence of this initial response.

Considering RT results, rather weak results were shown for Ltbp1 and Angpt2 at day 3. Results regarding day 7 are confusingly reported and the crucial comparison ASC vs saline is not reported in figure 5. Moreover, macrophage migration and positive signal transduction “related effects” seem to be too general and no data can support this hypothesis (macrophage infiltration analysis…?)

We agree regarding the day 7 data. We have described the results from day 7 differently now. The volcano plot from the comparison between ASC and saline group at day 7 has been added to the figure, while the plot for comparison between saline and ICM is moved to supplementary.

It is important to note, that the transcriptional results are derived from cardiac tissue in the peri-infarct region, while the flow cytometric data are derived from the whole heart. Therefore, the results will not be one-to-one. A local recruitment of macrophages from the other myocardial tissue to the peri-infarct region would only be apparent in the transcriptional data and not show in the flow cytometric data.

We have now disclaimed that the results from GeneOntology is based on relatively few differentially expressed genes, and that the observations should be addressed in future studies including methods such as infiltration analysis.

It is not clear how figure 6 was built and in my opinion, it appears as data repetition within the same work. Moreover, pie charts are not normally suggested for data comparison.

Figure 6 was not meant to represent new data, but rather to summarize the other observations in coherence to make it more readily available for the reader. We will make this clearer in the figure legend.

Reviewer 3 Report

This is a very well designed, written and comprehensive article. I only have some minor
comments.

1. The authors have mentioned in text lines 43-44 that “treatments using
mesenchymal stromal cells (MSCs) are the furthest in clinical translation, where the
first phase III study was recently concluded”. My suggestion is to support their
statement with some references of clinical trials. A helpful link is: clinicaltrials.gov

2. The authors quote the 2006 ISCT guidelines but these have been updated and
should be considered. A helpful reference is: Immunological characterization of
multipotent mesenchymal stromal cells--The International Society for Cellular
Therapy (ISCT) working proposal. (Cytotherapy 2013 Sep;15(9):1054-61.) Therefore,
it would be much better to put the latest and more relevant reference to support
their statement.

3. Figures 2-4. Symbols in graphs are too small. I suggest to simplify the gating strategy
in flow cytometry parts or to submit the whole strategy as supplementary material.

4. In line 132 is referred that “the percentage of the non-injected ICM group contained
roughly 9% CD45+, 68% CD31+CD45-, and 7% CD90+CDD45- cells”. Please check if
there is any wrong about the percentage of CD90+CDD45- cells. Accordingly, to
graph is over 8% almost 9%.

5. Figure 5. The information included in text does not justify the whole figure and vice
versa. For example, in line 242 is referred that colony stimulating factor 2, Vegfa and
Vegfb were significantly upregulated in the ASC group compared to ICM at day 7
after treatment. However, in the graph are presented only Vegfa (two times!) and
Vegfb, but nowhere is the “CSF-2”. The above comment is referred to figure 5C, but
in a same way please check also figure 5D. For example, please indicate in the graph
the presence of integrin subunit alpha V and kinase insert domain receptor.

Author Response

We thank the reviewer for his/her comments, and have answered them below: 

The authors have mentioned in text lines 43-44 that “treatments using mesenchymal stromal cells (MSCs) are the furthest in clinical translation, where the first phase III study was recently concluded”. My suggestion is to support their statement with some references of clinical trials. A helpful link is: clinicaltrials.gov

The NCT number (NCT02032004) has been added.

The authors quote the 2006 ISCT guidelines but these have been updated and should be considered. A helpful reference is: Immunological characterization of multipotent mesenchymal stromal cells--The International Society for Cellular Therapy (ISCT) working proposal. (Cytotherapy 2013 Sep;15(9):1054-61.) Therefore, it would be much better to put the latest and more relevant reference to support their statement.

Krampera, 2013 is now cited in the article.

Figures 2-4. Symbols in graphs are too small. I suggest to simplify the gating strategy in flow cytometry parts or to submit the whole strategy as supplementary material.

The images of the gating strategy are now made larger, which should make the text clearer.

In line 132 is referred that “the percentage of the non-injected ICM group contained roughly 9% CD45+, 68% CD31+CD45-, and 7% CD90+CDD45- cells”. Please check if there is any wrong about the percentage of CD90+CDD45- cells. Accordingly, to graph is over 8% almost 9%.

We completely agree with the reviewer, and this is a written mistake from our part. The graph is correct, and the text has accordingly been changed to 9 %.

  1. Figure The information included in text does not justify the whole figure and vice versa. For example, in line 242 is referred that colony stimulating factor 2, Vegfa and Vegfb were significantly upregulated in the ASC group compared to ICM at day 7 after treatment. However, in the graph are presented only Vegfa (two times!) and Vegfb, but nowhere is the “CSF-2”. The above comment is referred to figure 5C, but in a same way please check also figure 5D. For example, please indicate in the graph the presence of integrin subunit alpha V and kinase insert domain receptor.

The fact that VEGFa was written twice in the plot is a mistake. The plots are generated in Qiagens GeneGlobe software with no titles on the dots, thus they were added afterwards. The mistake has been corrected now.

Regarding KDR and ITGAV, they are already indicated in 5D.

Round 2

Reviewer 1 Report

The authors responded appropriately to comments and did all necessary changes to the text and figures.  Their responses are really appreciated.

Author Response

We thank the reviewer for his/her quick and positive response and the overall interaction

Reviewer 2 Report

Distinguished Authors,

I carefully read the resubmitted version of the article ”The immediate cardiac tissue response to cryopreserved allogeneic adipose tissue-derived mesenchymal stromal cells in rats with chronic ischemic cardiomyopathy.” by Bjarke Follin et al..

I appreciate title editing and efforts to ameliorate the work.

I think most of my observations were taken in consideration and, at least a major part of them, led to text changes.

I still have few issues puzzling me.

In results section, the sentence at line 177 ”The response would then follow the cardiac tissue response to insults such as myocardial infarction” sounds not clear: please try to reformulate it.

As stated, ASC were thawed and infused in presence of the criopreserving medium itself. Can you explain why did you use saline as control injections instead of the cryopreserving medium CryoStor10.

Administration of thawed ASC was chosen to closely resemble the scenario of clinical applications. Nevertheless, cells were expanded in presence of FBS, a medium additive that is not compatible with GMP compliant ex vivo cell expansion protocols. Please discuss.

Author Response

We thank the reviewer for his/her quick and positive response.

We have addressed the questions and corrected minor details in the manuscript. We hope that these additions to the manuscript will make the messages more clear.

In results section, the sentence at line 177 ”The response would then follow the cardiac tissue response to insults such as myocardial infarction” sounds not clear: please try to reformulate it.

We agree that this formulation could be improved. Our new formulation is added:

“The timing of the response seems similar to other cardiac wound healing timelines, eg. after myocardial infarction.”

As stated, ASC were thawed and infused in presence of the criopreserving medium itself. Can you explain why did you use saline as control injections instead of the cryopreserving medium CryoStor10.

This is to keep the design of the study as close to the clinical studies as possible. In those studies, the placebo group receives saline, and not the CryoStor10. In the section where we first describe the injections, we have now written:

“The choice of saline as the placebo group was to keep the design as close to the clinical studies.”

Administration of thawed ASC was chosen to closely resemble the scenario of clinical applications. Nevertheless, cells were expanded in presence of FBS, a medium additive that is not compatible with GMP compliant ex vivo cell expansion protocols. Please discuss.

We have tried expanding rat ASCs in growth medium similar to the one used for GMP expansion. However, these was no added benefit in proliferation, as observed with the human ASCs, indicating that the supplement does not affect the rat ASCs similarly to the human ASCs. In some cases, we in fact observed cell death (data not published).

“The choice of FBS as expansion supplement was made after comparing it with GMP compliant supplements on rat ASCs. We found that the rat ASCs did not respond to the GMP compliant supplement in the same way as the human ASCs.”